# Sperm competition risk drives rapid ejaculate adjustments mediated by seminal fluid

Michael J Bartlett[1]*, Tammy E Steeves[1], Neil J Gemmell[2], Patrice C Rosengrave[2]

[1]School of Biological Sciences, University of Canterbury, Christchurch, New Zealand; [2]Department of Anatomy, University of Otago, Dunedin, New Zealand

**Abstract** In many species, males can make rapid adjustments to ejaculate performance in response to sperm competition risk; however, the mechanisms behind these changes are not understood. Here, we manipulate male social status in an externally fertilising fish, chinook salmon (*Oncorhynchus tshawytscha*), and find that in less than 48 hr, males can upregulate sperm velocity when faced with an increased risk of sperm competition. Using a series of *in vitro* sperm manipulation and competition experiments, we show that rapid changes in sperm velocity are mediated by seminal fluid and the effect of seminal fluid on sperm velocity directly impacts paternity share and therefore reproductive success. These combined findings, completely consistent with sperm competition theory, provide unequivocal evidence that sperm competition risk drives plastic adjustment of ejaculate quality, that seminal fluid harbours the mechanism for the rapid adjustment of sperm velocity and that fitness benefits accrue to males from such adjustment.
DOI: https://doi.org/10.7554/eLife.28811.001

*For correspondence:
michael.bartlett@pg.canterbury.
ac.nz

Competing interests: The authors declare that no competing interests exist.

## Introduction

Sperm competition (*Parker, 1970*) occurs commonly across many invertebrate and vertebrate taxa and is a potent evolutionary force influencing male reproductive biology (*Birkhead and Møller, 1998*; *Birkhead and Pizzari, 2002*; *Simmons and Fitzpatrick, 2012*). Sperm competition theory predicts that males will trade-off between energy expended making high-quality ejaculates with obtaining mating opportunities and that males will invest differentially in ejaculates with respect to sperm competition risk (*Parker, 1990*; *Parker et al., 1997*; *Parker, 1998*; *Wedell et al., 2002*; *Birkhead et al., 2009*; *Parker and Pizzari, 2010*). In agreement with these predictions, males of many species can make rapid adjustments to ejaculate quality within days (*Rudolfsen et al., 2006*; *Pizzari et al., 2007*; *Thomas and Simmons, 2007*; *Gasparini et al., 2009*; *Smith and Ryan, 2011*), hours (*Cornwallis and Birkhead, 2007a*) and even minutes (*Kilgallon and Simmons, 2005*; *Joseph et al., 2015*) of exposure to a new social cue that signals changing sperm competition risk, such as the presence of a female, or a male competitor. For example, in fowl (*Gallus gallus*), males of dominant social status strategically allocate sperm, ejaculating more and faster swimming sperm in initial copulations and to females of higher quality (*Pizzari et al., 2003*; *Cornwallis and Birkhead, 2006*; *Cornwallis and Birkhead, 2007a*; *Cornwallis and Birkhead, 2007b*), and alter their allocation strategy accordingly when changing social status (*Cornwallis and Birkhead, 2007a*). While males of several vertebrate species ranging from fish (*Rudolfsen et al., 2006*; *Gasparini et al., 2009*; *Smith and Ryan, 2011*) to humans (*Kilgallon and Simmons, 2005*; *Joseph et al., 2015*) can strategically alter the quality of their ejaculate in response to social cues, the mechanism behind such rapid adjustments is as yet unknown.

A promising candidate mechanism for rapid adjustment of sperm velocity may be found in the non-sperm component (seminal fluid and its constituents) of the ejaculate, particularly if such

**eLife digest** Males of many animal species fight to establish social dominance and control access to females so that they have more opportunities to reproduce than their competitors. Males with lower social status will struggle to directly compete for mates, thus they attempt to mate with females by stealth. This often leads to more than one male mating with the same female so that the sperm from each male end up competing to fertilise that female's eggs, a phenomenon known as sperm competition.

Males suspend their sperm in a fluid to make a mixture known as semen. It has been shown that, compared to high status males, low status males will produce higher quality semen that contains greater numbers of faster swimming sperm, giving them an advantage in sperm competition. Growing evidence from several species indicates that males can quickly adjust how fast their sperm swim in response to social cues that signal changing risks of sperm competition. However, how these rapid adjustments occur remains largely unknown, and whether they alter a male's reproductive success against a competitor has seldom been examined.

Chinook salmon usually live in the North Pacific Ocean but they swim up rivers in North America and Asia to reproduce. They have also been introduced to several other countries including New Zealand where they are farmed commercially. The fish are highly prized by sport fishermen and are also of cultural significance to certain groups of indigenous people in North America. Barlett et al. studied the semen of chinook salmon, undertaking a series of experiments in which males switched between high and low social status. The experiments show that, as predicted, the sperm of males that changed from high to low social status started to swim faster. These changes in speed were caused by the fluid in the semen and altered the number of eggs that the male's sperm fertilised when competing against sperm from another male.

In their natural range some populations of chinook salmon are declining due to overfishing combined with habitat loss and alteration. The findings of Barlett et al. contribute to a better understanding of how this fish species reproduces, which may lead to the introduction of measures that help natural populations to recover or help to improve commercial farming. Improved knowledge of how the fluid in semen affects sperm activity may also have important consequences for our wider understanding of male fertility in humans and other animals.
DOI: https://doi.org/10.7554/eLife.28811.002

adjustments occur more rapidly than spermatogenesis (*Cameron et al., 2007*; *Perry et al., 2013*; *Fitzpatrick and Lüpold, 2014*). Seminal fluid is a complex medium containing a great diversity of molecules (*Poiani, 2006*; *Juyena and Stelletta, 2012*) and is known to influence sperm velocity and motility in vertebrates (*Lahnsteiner et al., 1998*; *Lahnsteiner et al., 1996*; *Poiani, 2006*; *Locatello et al., 2013*; *González-Cadavid et al., 2014*). For example, research using an externally fertilising fish, the grass goby (*Zosterisessor ophiocephalus*), compared males for which sperm competition strategy is determined by age/size and found large males that adopt a guarding strategy have a greater concentration of the seminal fluid glycoprotein mucin (*Scaggiante et al., 1999*). Furthermore, by separating and recombining seminal fluid and sperm from different males, research using the same species found seminal fluid had a tactic-specific effect on sperm velocity, with seminal fluid from sneak males decreasing the velocity of rival guard male sperm and seminal fluid from guard males increasing the velocity of sneak male sperm (*Locatello et al., 2013*).

However, only one study to date has investigated the role that seminal fluid plays as a mediator of short-term plastic sperm performance in a vertebrate species using fowl and the results were inconsistent with theoretical expectation: *Cornwallis and O'Connor, 2009* found that while ejaculates produced by male fowl that were allocated to females of higher quality contained faster sperm, seminal fluid from those ejaculates reduced the velocity of sperm from the same male allocated to females of lower quality. To be consistent with the prediction that seminal fluid mediates changes in sperm velocity, seminal fluid from ejaculates allocated to higher quality females should increase, not decrease the speed of sperm isolated from ejaculates allocated to lower quality females. Thus, although there is evidence that seminal fluid can influence sperm velocity, evidence that seminal fluid

mediates the rapid plastic adjustment of an ejaculate's motile performance consistent with theoretical expectation is lacking.

We use an ideal model species, chinook salmon (*Oncorhynchus tshawytscha*), to examine patterns of ejaculate plasticity in response to changes in male social status and the reproductive consequences of these changes. In salmonids, fertilisation occurs externally and sperm competition occurs in the majority of spawnings (*Berejikian et al., 2010*; *Sørum et al., 2011*). Male chinook salmon adopt Alternative Reproductive Tactics (ARTs) situationally, as 'hooknose' males fight to establish social dominance (*Esteve, 2005*). Only dominant males guard spawning females thus obtaining priority in mating position, while subdominant males that lose contests attempt to sneak fertilisations by invading spawning pairs and releasing their sperm (*Esteve, 2005*). The social status of male salmon is subject to change over the course of a spawning season; for example, in coho salmon (*O. kisutch*), 22% of observed contests between hooknose males resulted in displacement of the previous dominant male (*Healey and Prince, 1998*). Therefore, in this mating system, females mate with multiple males in a dynamic social environment that results in intense levels of fluctuating sperm competition risk.

Previous research has shown that when males engage in sperm competition, sperm swimming speed is the primary predictor of fertilisation success in chinook salmon (*Evans et al., 2013*; *Rosengrave et al., 2016*) and other salmonids (*Gage et al., 2004*; *Liljedal et al., 2008*; *Egeland et al., 2015*). Sperm competition theory therefore predicts subdominant males, which have greater sperm competition risk, will invest in ejaculates with faster swimming sperm than dominant males and males changing social status should adjust their investment accordingly (*Parker, 1990*; *Parker et al., 1997*; *Parker, 1998*; *Wedell et al., 2002*; *Birkhead et al., 2009*; *Parker and Pizzari, 2010*). Indeed, several studies that experimentally manipulated social status using Arctic charr (*Salvelinus alpinus*) have found that subdominant males produce ejaculates with more sperm and faster swimming sperm than dominant males (*Liljedal and Folstad, 2003*; *Rudolfsen et al., 2006*; *Vaz Serrano et al., 2006*; *Haugland et al., 2009*). Furthermore, *Rudolfsen et al. (2006)* demonstrated that following a social challenge, both sperm concentration and velocity decreased over a 4-day period compared with pre-trial levels in dominant males, and observed an increase in sperm concentration but no change in sperm velocity for subdominant males. However, *Rudolfsen et al. (2006)* did not evaluate male social status prior to the social challenge, so it is unknown if these males actually changed or simply retained the same status through the course of the experiment. Recent research shows that ejaculates from subdominant Arctic charr sire the same number of eggs when in competition with ejaculates from dominant males if their sperm were released after the average delay observed under natural conditions (*Egeland et al., 2015*). These results suggest that salmonid males in disfavoured mating positions can compensate by producing more competitive ejaculates than dominant males, but whether males changing social status adjust their sperm velocity, and if such adjustments to ejaculates are mediated by sperm or non-sperm components of the ejaculate, is yet to be determined.

Here, we use a comprehensive experimental approach to determine if changes in sperm velocity observed in response to an individual's social position are the result of alterations to the gametes or to seminal fluid and if such responses actually alter a male's reproductive success against a sperm competitor. Specifically, we examine whether ejaculate quality is phenotypically plastic in response to changes in sperm competition risk over 48 hr periods, using a two-stage challenge to manipulate social status (*Cornwallis and Birkhead, 2007a*; *Pizzari et al., 2007*) and collected ejaculates at each stage of the experiment. In the second stage, males either retained or were forced to change their social status, creating four social phenotypes with varying sperm competition risk (*Figure 1*). We found that subdominant males, which have greater sperm competition risk, invest more in both sperm concentration and sperm velocity compared to socially dominant males. Additionally, we find males that change from dominant to subdominant social status, thus elevated their sperm competition risk, increased their sperm velocity as predicted by sperm competition theory (*Parker, 1990*; *Parker et al., 1997*; *Parker, 1998*; *Wedell et al., 2002*; *Birkhead et al., 2009*; *Parker and Pizzari, 2010*). We also separated sperm from seminal fluid and created reciprocal combinations both within and between rival males, finding that males can make rapid adjustments to sperm velocity by producing seminal fluid that enhances sperm function. We then used *in vitro* fertilisation trials and found the seminal fluid effects on sperm swimming speed influences male reproductive success under sperm competition. Our combined experimental results provide compelling evidence that seminal fluid is the mediator of rapid strategic adjustment of sperm velocity, thus bringing us a critical step

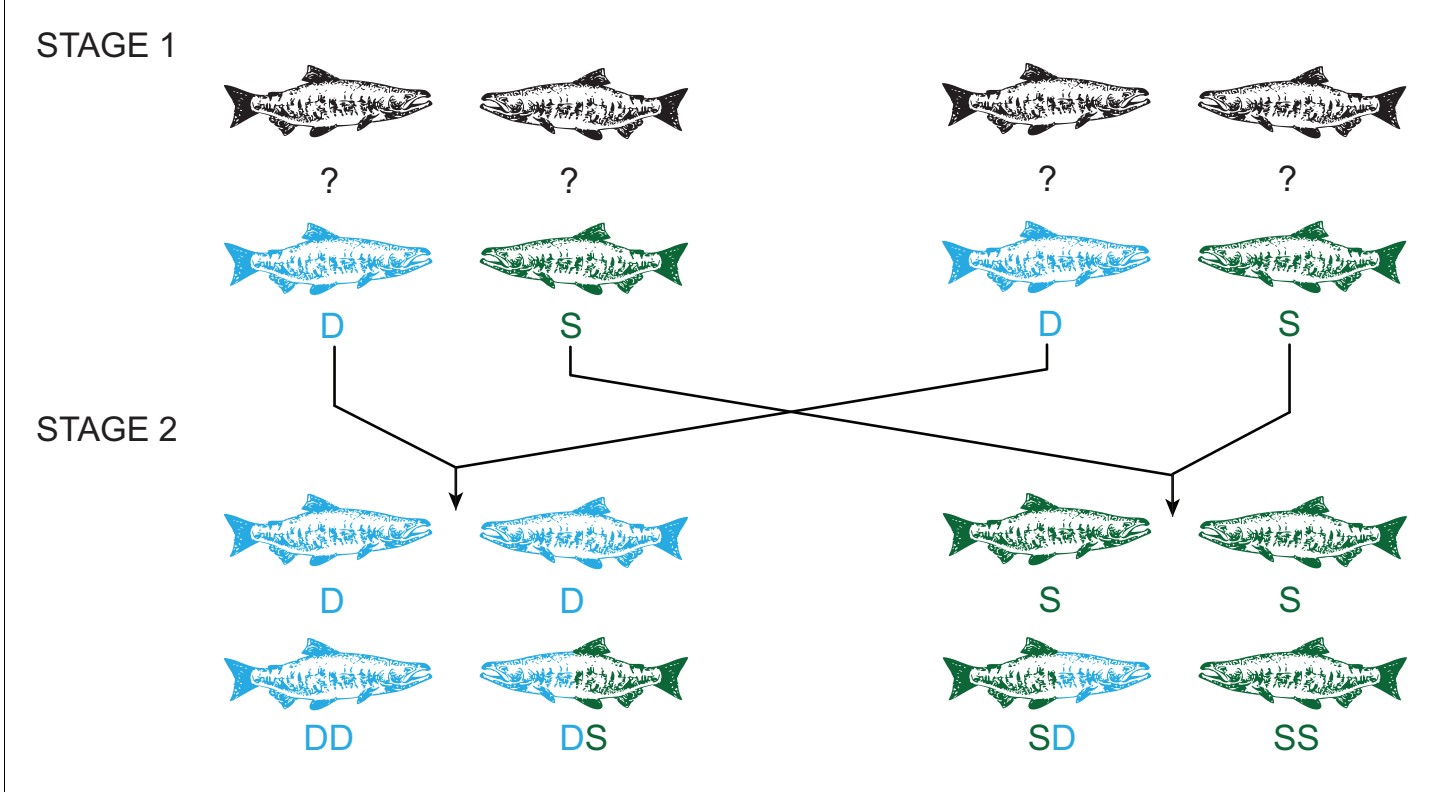

**Figure 1.** Experimental design using a two-stage social status manipulation in chinook salmon. For each trial, in stage 1, four males of unknown social status were used to form two pairs and the social hierarchy within each pairing was then determined, assigning one male as dominant (D) and the other subdominant (S). After 48 hr, ejaculates were collected from each male (D, S, D, S). In stage 2, we reformed pairs, putting males with the same social status together, and re-determined the social hierarchy within each pairing. Males either retained the same status, dominant (DD) or subdominant (SS) in both stages, or changed status in either direction, dominant to subdominant (DS) or subdominant to dominant (SD). After 48 hr, ejaculates were recollected from each male (DD, DS, SD, SS).
DOI: https://doi.org/10.7554/eLife.28811.003

closer to identifying the underlying molecular mechanism that enables plasticity of ejaculate performance in dynamic social environments.

## Results

### Social status and ejaculate quality

Subdominant (S) males had on average faster swimming sperm (Average Path Velocity, or VAP) than dominant (D) males. This difference was not significant when social status was initially determined in stage 1 (*Table 1*; *Figure 2a*) but was significant for stage 2 (*Table 1*; *Figure 2b*). Overall, there was considerable variation in sperm swimming speeds among males, accounted for by the random predictor 'male identity' that was significant in both stages (stage 1: $\chi^2_{(1)}$=105.11, p<0.001; stage 2: $\chi^2_{(1)}$=70.02, p<0.001). Additionally, sperm concentration was significantly higher in S than in D males in stage 1 (*Table 1*; *Figure 3a*), but not stage 2 (*Table 1*; *Figure 3b*). However, sperm concentration for males that remained subdominant (SS) was significantly higher than for those males that remained socially dominant (DD) in stage 2 (*Table 1*).

### Ejaculate plasticity in response to social status change

There was a significant increase in mean VAP for males that changed from dominant to subdominant social status (DS; *Table 2*; *Figure 4*). Throughout the social status experiment, there were no other significant changes in either VAP or sperm concentration for males of the other social phenotypes (*Table 2*; *Figure 4*). There was also a significant overall interaction effect between social phenotype

**Table 1.** Generalised linear mixed effects models (GLMM) to compare sperm velocity (VAP, μs$^{-1}$) and sperm concentration (cells/ml) among male chinook salmon of different social status (see *Figure 1* for experimental design).

In stage 1 of the experiment, dominant (D; n = 22) males were compared to subdominants (S; n = 22). In stage 2, separate models were run with the fixed parameter social status with either four levels (males that retained the same status DD (n = 10) and SS (n = 9), and males that changed status SD (n = 9) and DS [n = 10]), or two levels with data pooled together (D = DD + SD (n = 19), S = SS + DS (n = 19)). p-Values are calculated using Satterthwaite approximations to degrees of freedom and 95% confidence intervals were calculated using the Wald method. p-Values are adjusted for multiple testing where multiple pairwise comparisons are made using the Bonferroni method with significant values highlighted in bold.

| Response variable | Stage | Parameters (fixed effects) | Estimate | 95% CI | p Value |
|---|---|---|---|---|---|
| VAP | 1 | Intercept | 152.9 | 135.3–170.4 | |
| | | D – S | 7.4 | −8.6–23.4 | 0.37 |
| | 2 | Intercept | 127.1 | 108.8–145.5 | |
| | | D – S | 19.7 | 5.1–34.2 | **0.01** |
| | 2 | Intercept | 131.2 | 109.2–153.2 | |
| | | DD – SS | 14.9 | −6.5–36.5 | 0.18 |
| | | DD – DS | 17.9 | −2.7–38.5 | 0.09 |
| | | SD – DS | 24.4 | 2.9–45.9 | 0.03 |
| | | SD – SS | 21.5 | −0.2–43.2 | 0.06 |
| Sperm concentration | 1 | Intercept | 6.0 | 5.81–6.22 | |
| | | D – S | 0.2 | 0.01–0.39 | **0.04** |
| | 2 | Intercept | 5.9 | 5.72–6.21 | |
| | | D – S | 0.2 | −0.06–0.41 | 0.14 |
| | 2 | Intercept | 5.8 | 5.55–6.09 | |
| | | DD – SS | 0.5 | 0.16–0.77 | **0.003** |
| | | DD – DS | 0.1 | −0.16–0.43 | 0.36 |
| | | SD – DS | −0.1 | −0.44–0.18 | 0.42 |
| | | SD – SS | 0.2 | −0.12–0.52 | 0.21 |

DOI: https://doi.org/10.7554/eLife.28811.012

and experimental stage on VAP ($\chi^2_{(3)}$=11.8, p=0.008), with a significant interaction effect found only for males changing from dominant to subdominant status (DS; p=0.02, 95% CI = 2.9–34.9). We found no significant interaction effects between social phenotype and experimental stage on sperm concentration ($\chi^2_{(3)}$=3.0, p=0.385).

## Seminal fluid effect on sperm velocity

Within each dyad, the social status of the rival male was a significant predictor of the difference in VAP between focal male's sperm incubated in their own seminal fluid and the focal male's sperm incubated in their rival's seminal fluid (*Table 3*). Seminal fluid from subdominant males increased the sperm swimming speed of sperm from dominant males, conversely, the seminal fluid from dominant males decreased sperm swimming speed of the sperm from subdominant males (*Figure 5*). However, rival's social status was no longer significant (*Table 3*) when the difference in VAP between the focal male control and rival male control was added as a fixed predictor to the model, for which a significant positive linear relationship was detected (*Table 3*), with sperm in the seminal fluid of a rival that had faster VAP increasing sperm velocity and sperm in the seminal fluid of a rival that had slower VAP decreasing sperm velocity relative to VAP in their own seminal fluid (*Figure 6*).

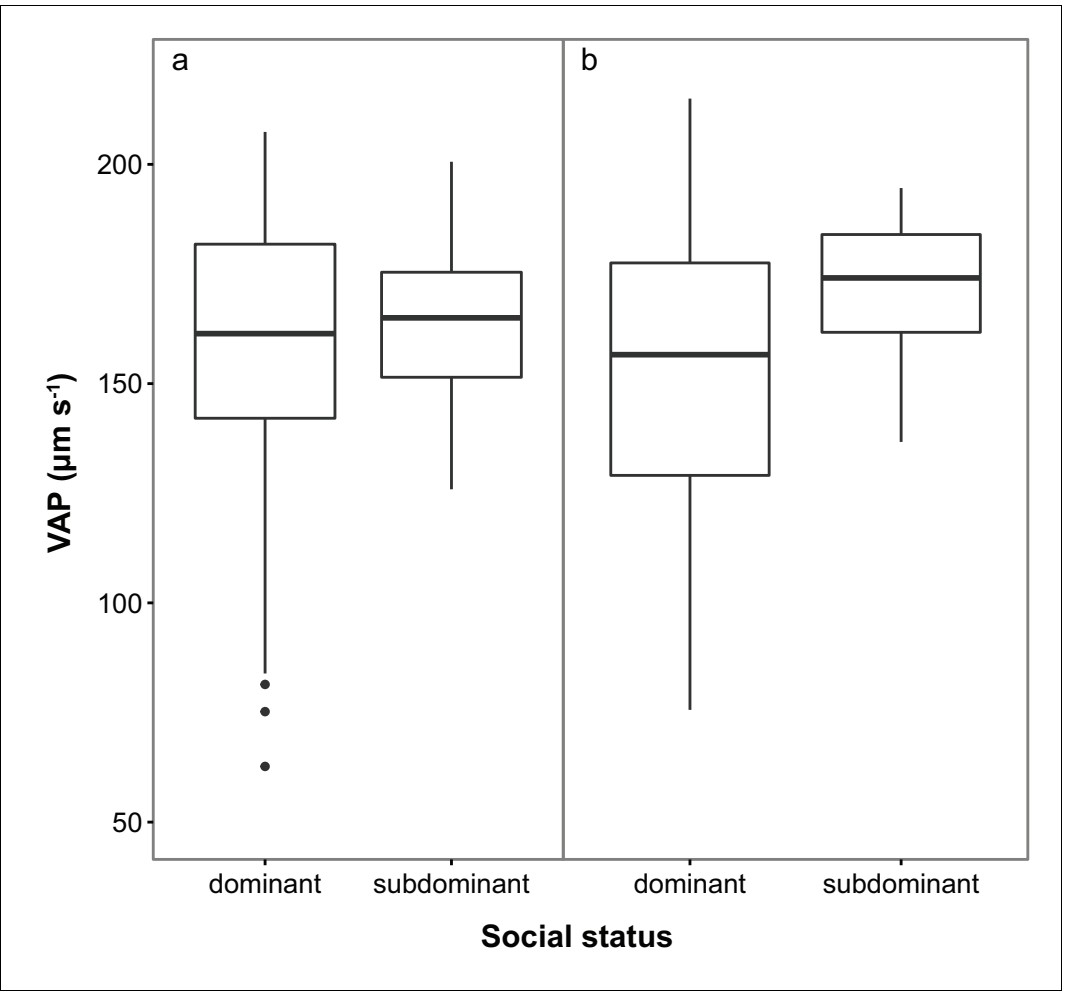

**Figure 2.** Sperm velocity (VAP in µm s$^{-1}$) in males of dominant (D) and subdominant (S) social status after a: the first social challenge (D, n = 22; S, n = 22) and b: the second social challenge (D, n = 19; S, n = 19). Boxplots display the median of each group with the 25th and 75th percentiles and whiskers extend to data within 1.5 x the inter-quartile range.

DOI: https://doi.org/10.7554/eLife.28811.004

The following source data and figure supplements are available for figure 2:

**Source data 1.** Source data for boxplot (*Figure 2a*).
DOI: https://doi.org/10.7554/eLife.28811.006
**Source data 2.** Source data for boxplot (*Figure 2b*).
DOI: https://doi.org/10.7554/eLife.28811.007
**Figure supplement 1.** Across all sperm samples collected in this study, Average Path Velocity (VAP) at 10 s post-activation was strongly correlated with Curvilinear Velocity (VCL; r = 0.85, p<0.000 l, n = 126).
DOI: https://doi.org/10.7554/eLife.28811.005
**Figure supplement 1—source data 1.** Source data for correlation analysis.
DOI: https://doi.org/10.7554/eLife.28811.008

### *In vitro* fertilisation trials

Male social status was a significant predictor of the proportion of eggs sired (*Table 4*), with subdominant males siring a greater proportion (0.54 ± 0.08 95% CI, n = 21) than dominant males (0.46 ± 0.06 95% CI, n = 21). The social status of the seminal fluid donor when seminal fluid was swapped between males was also a significant predictor of the proportion of eggs sired (*Table 4*), with sperm incubated in the seminal fluid of subdominant males siring a greater proportion (0.6 ± 0.09 95% CI, n = 21) of eggs than sperm incubated in the seminal fluid of dominant males (0.4 ± 0.09 95% CI,

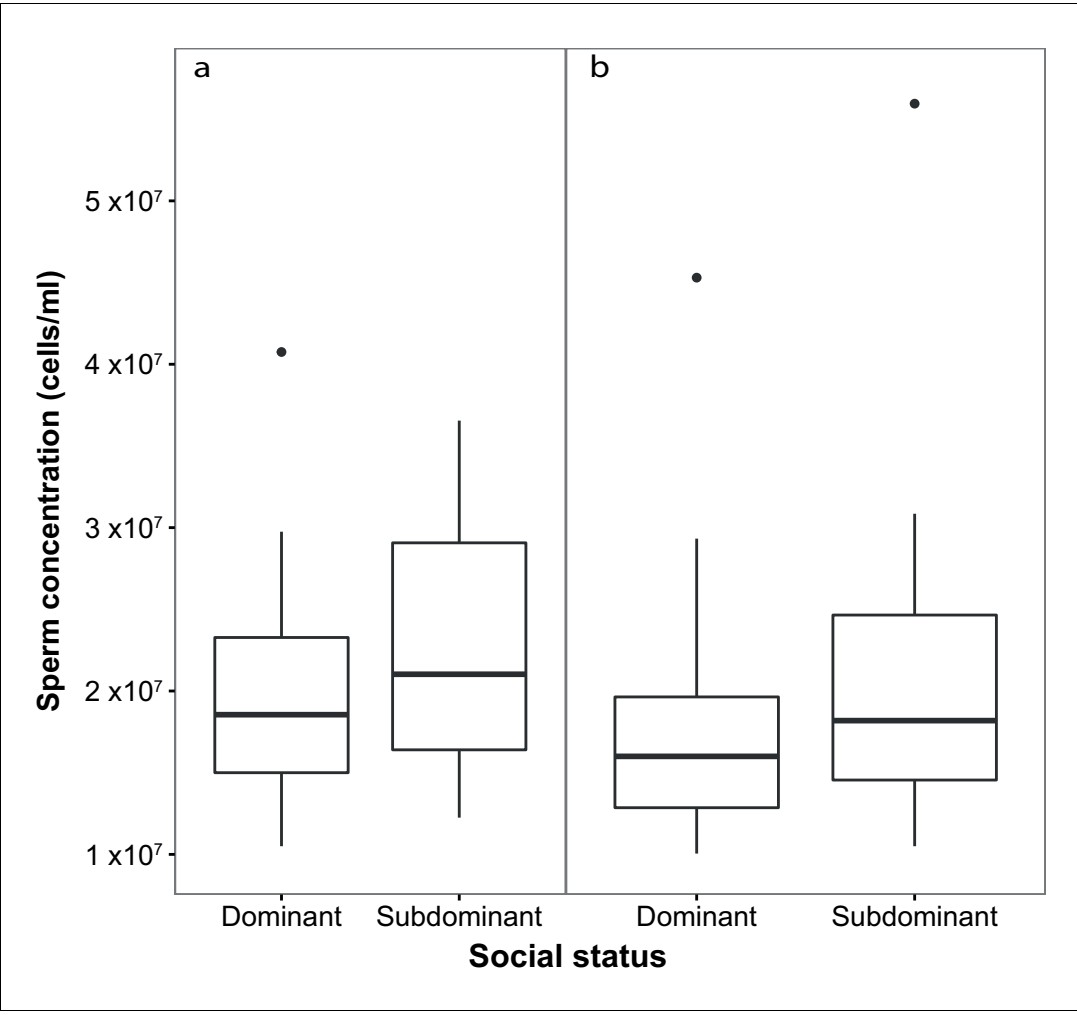

**Figure 3.** Sperm concentration (cells/ml) in males of dominant (D) and subdominant (S) social status after a: the first social challenge (D, n = 22; S, n = 22) and b: the second social challenge (D, n = 19; S, n = 19). Boxplots display the median of each group with the 25th and 75th percentiles and whiskers extend to data within 1.5 x the inter-quartile range.

DOI: https://doi.org/10.7554/eLife.28811.009

The following source data is available for figure 3:

**Source data 1.** Source data for boxplot (*Figure 3a*).
DOI: https://doi.org/10.7554/eLife.28811.010
**Source data 2.** Source data for boxplot (*Figure 3b*).
DOI: https://doi.org/10.7554/eLife.28811.011

n = 21). The difference in sperm velocity between competitors was also a significant predictor of the proportion of eggs sired in both unmanipulated (*Table 4*) and recombined ejaculate seminal fluid treatments (*Table 4*). The change in relative sperm velocity between males within the same male-male-female combinations across seminal fluid treatments was a significant predictor of the change in the proportion of eggs sired by that male's sperm across treatments (*Table 4*; *Figure 7*).

## Discussion

In this study, we experimentally manipulated social status to produce four social phenotypes with differing levels of sperm competition risk, and in accordance with sperm competition theory (*Parker, 1990*; *Parker et al., 1997*; *Parker, 1998*; *Wedell et al., 2002*; *Birkhead et al., 2009*; *Parker and Pizzari, 2010*), found males with the highest risk of sperm competition produced

**Table 2.** Generalised linear mixed effects models (GLMM) to compare sperm velocity (VAP, μs$^{-1}$) and sperm concentration (cells/ml) in males of each social phenotype changing from stage 1 to stage 2 of the experiment.

The four social phenotypes are males that remained dominant (DD, n = 10) or subdominant (SS, n = 9) in both stages and males that changed status in either direction, subdominant to dominant (SD, n = 9) and dominant to subdominant (DS, n = 10). p-Values are calculated using Satterthwaite approximations to degrees of freedom and 95% confidence intervals were calculated using the Wald method. p-Values are adjusted for multiple testing using the Bonferroni method with significant values highlighted in bold.

| Response variable | Social phenotype | Parameters (fixed effects) | Estimate | 95% CI | p alue |
|---|---|---|---|---|---|
| VAP | DD | Intercept | 109.1 | 88.9–129.2 | |
| | | Stage 1 – Stage 2 | 0.1 | −14.1–14.4 | 0.9 |
| | SD | Intercept | 139.6 | 111.9–167.2 | |
| | | Stage 1 – Stage 2 | −8.9 | −19.5–1.5 | 0.1 |
| | DS | Intercept | 163.9 | 141.1–186.8 | |
| | | Stage 1 – Stage 2 | 17.2 | 5.4–29.1 | **0.008** |
| | SS | Intercept | 162.5 | 147.1–177.9 | |
| | | Stage 1 – Stage 2 | −2.3 | −12.0–7.4 | 0.6 |
| Sperm concentration | DD | Intercept | 5.6 | 5.34–5.97 | |
| | | Stage 1 – Stage 2 | −0.2 | −0.39–0.06 | 0.2 |
| | SD | Intercept | 6.4 | 6.12–6.68 | |
| | | Stage 1 – Stage 2 | −0.2 | −0.48–0.002 | 0.05 |
| | DS | Intercept | 6.1 | 5.56–6.58 | |
| | | Stage 1 – Stage 2 | −0.1 | −0.34–0.15 | 0.4 |
| | SS | Intercept | 6.4 | 6.09–6.61 | |
| | | Stage 1 – Stage 2 | 0.1 | −0.17–0.35 | 0.5 |

DOI: https://doi.org/10.7554/eLife.28811.016

ejaculates with both higher sperm concentration and faster swimming sperm. We also found males can make rapid adjustments to sperm velocity in a strategic response to changes in social position that signal increased sperm competition risk. While seminal fluid is often *implicated* to harbour the unknown mechanism behind plastic sperm performance in vertebrates (*Perry et al., 2013*; *Fitzpatrick and Lüpold, 2014*), our combined results for the first time, unequivocally demonstrate that seminal fluid acts as a mediator of rapid strategic adjustment to sperm velocity. Furthermore, we demonstrate strategic adjustments of sperm velocity mediated by seminal fluid directly impact male fitness, highlighting the adaptive significance of plastic ejaculate performance.

Sperm competition theory predicts that males should strategically adjust ejaculates in response to changing sperm competition risk (*Wedell et al., 2002*; *Parker and Pizzari, 2010*). In chinook salmon, relative sperm velocity among males is the primary determinant of fertilisation success (*Evans et al., 2013*; *Rosengrave et al., 2016*). We show males forced to change from dominant to subdominant social status, and therefore exposed to increased sperm competition risk, responded by increasing the quality of their ejaculate, in this case sperm velocity, within 48 hr (*Figure 4*). While we predict that males forced to change from subdominant to dominant social status, therefore exposed to decreased sperm competition risk, would respond by decreasing their ejaculate quality, we did not see a significant change in sperm velocity for these males. However, subdominant males that later became dominant had a relatively low mean sperm velocity that was more similar to dominant males than those from the other subdominant phenotype in the first stage of the experiment (*Figure 4*). In this case, these subdominant males may have attempted to adopt a guarding tactic even after losing in the first social challenge, as males that lose contests can either sneak or fight for dominance elsewhere (*Esteve, 2005*).

Males should also strategically adjust sperm concentration in response to changing sperm competition risk (*Wedell et al., 2002*; *Parker and Pizzari, 2010*). Accordingly, we found subdominant males produced ejaculates with greater sperm concentration than dominant males. However, our

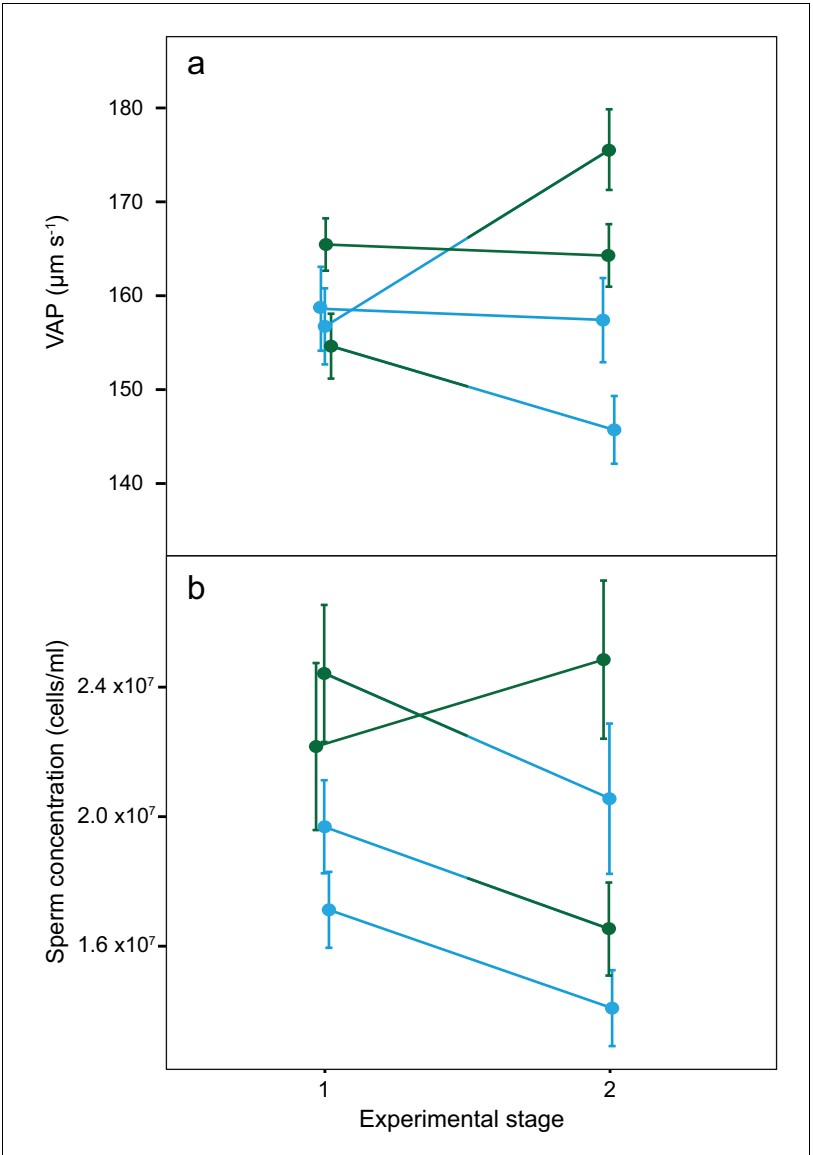

**Figure 4.** Average sperm velocity (VAP, μm s$^{-1}$;±s.e.m.) and average sperm concentration (cells/ml;±s.e.m.) in males of the four social phenotypes after each stage of a social status manipulation experiment in chinook salmon. Blue colour denotes males dominant in both stages (DD, n = 10), green colour denotes males subdominant in both stages (SS, n = 9), a change from blue to green colour denotes males that changed from dominant to subdominant status (DS, n = 10) and a change from green to blue colour denotes males that changed from subdominant to dominant status (SD n = 9). The change in VAP for DS males was statistically significant.
DOI: https://doi.org/10.7554/eLife.28811.013

The following source data is available for figure 4:

**Source data 1.** Source data for *Figure 4a*.
DOI: https://doi.org/10.7554/eLife.28811.014
**Source data 2.** Source data for *Figure 4b*.
DOI: https://doi.org/10.7554/eLife.28811.015

results show that there was no significant increase in sperm concentration for any of the social phenotypes over a 48-hr period. The exact time taken for spermatogenesis in salmonids is unknown; however, the process almost certainly takes more than 48 hr (*Billard, 1983a*; *Billard, 1983b*; *Schulz et al., 2010*). Therefore, these results suggest that the observed changes in sperm velocity

**Table 3.** Generalised linear mixed effects models (GLMM) predicting the change in sperm velocity (VAP, μs$^{-1}$) observed in the focal male's sperm when incubated in either their own seminal fluid or the seminal fluid of their rival male in that dyad, using the social status of the rival male and the relative VAP between sperm from focal and rival males as measured in their own seminal fluid (n = 42 males in 39 dyads).

$p$-Values are calculated using Satterthwaite approximations to degrees of freedom and 95% confidence intervals were calculated using the Wald method. Significant values are highlighted in bold.

| Response variable | Model | Parameters (fixed effects) | Estimate | 95% CI | p-Value |
|---|---|---|---|---|---|
| Change in VAP | 1 | Intercept | −24.4 | −41.8−−7.0 | |
| | | Rival's Social Status | 31.4 | 15.1–47.7 | **0.0003** |
| | 2 | Intercept | −0.64 | | |
| | | Rival's Social Status | 0.44 | −0.7–1.6 | 0.465 |
| | | Relative VAP | 0.05 | 0.04–0.07 | **<0.0001** |

DOI: https://doi.org/10.7554/eLife.28811.020

are mediated by a component of the ejaculate that modifies the competitiveness of existing sperm, rather than simply via the production of new sperm.

Our results clearly demonstrate the observed plasticity of sperm velocity in chinook salmon, a key determinant of fertilisation success in several vertebrate species (*Birkhead et al., 1999*; *Malo et al., 2005*; *Gasparini et al., 2010*; *Boschetto et al., 2011*) including salmonids (*Gage et al., 2004*; *Liljedal et al., 2008*; *Evans et al., 2013*; *Egeland et al., 2015*; *Rosengrave et al., 2016*), is mediated by seminal fluid. We found sperm from the same male, when incubated in seminal fluid from different males, had significantly different sperm velocities, and the direction of this effect could be predicted by social status. For example, when sperm from dominant males were incubated in seminal fluid from subdominant males we found that on average their sperm velocity increased

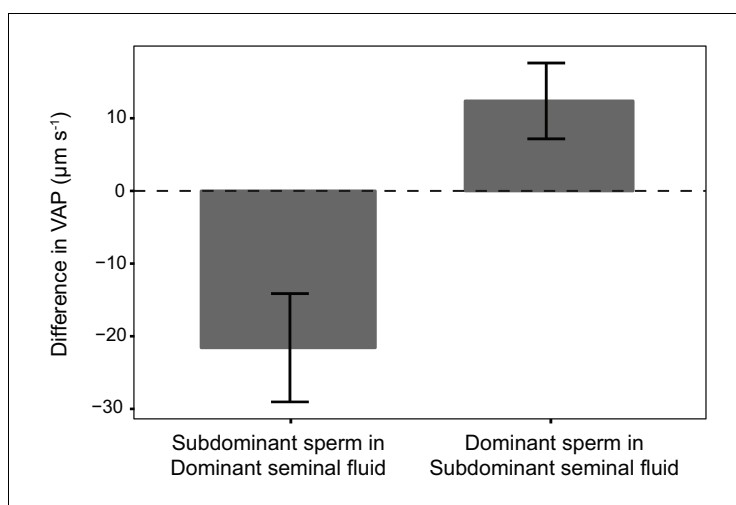

**Figure 5.** Average difference in sperm velocity (VAP, μm s$^{-1}$;±s.e.m.) between sperm incubated in their own seminal fluid and incubated in the seminal fluid of their rival in each dyad of a social status manipulation experiment in chinook salmon (n = 42 males in 39 dyads). Seminal fluid from dominant rival males on average decreased VAP of sperm from subdominant males. In contrast, seminal fluid from rival subdominant males on average increased VAP of sperm from dominant males. Social status was a significant predictor of the difference in sperm velocity between sperm incubated in their own seminal fluid and incubated in the seminal fluid of their rival.
DOI: https://doi.org/10.7554/eLife.28811.017
The following source data is available for figure 5:

**Source data 1.** Source data for *Figure 5*.
DOI: https://doi.org/10.7554/eLife.28811.018

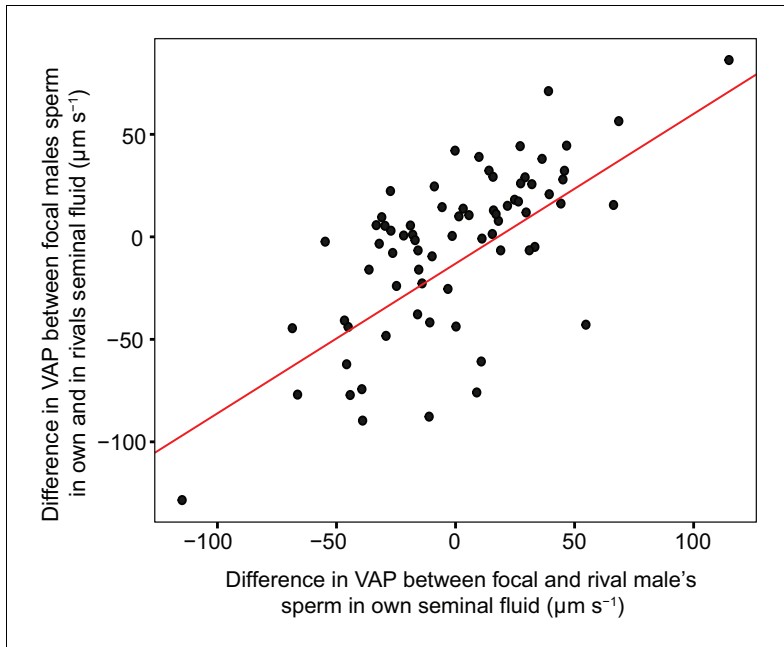

**Figure 6.** Significant linear relationship between the difference in sperm velocity (VAP, $\mu$m s$^{-1}$), between sperm incubated in their own seminal fluid and incubated in the seminal fluid of their rival, and the difference in VAP between sperm from the males in each pairing incubated in their own seminal fluid for each dyad of a social status manipulation experiment in chinook salmon (n = 42 males in 39 dyads). Incubating sperm in the seminal fluid of a rival with faster VAP generally results in an increase in that male's sperm velocity. Likewise, incubating sperm in the seminal fluid of a rival with slower VAP generally results in a decrease in that male's sperm velocity. Raw data is displayed for ease of interpretation, data analysis required transformation (refer to Materials and methods: Statistical analyses and supplementary material for details).

DOI: https://doi.org/10.7554/eLife.28811.019

compared to the baseline measures in their own seminal fluid, and found the opposite effect when sperm from subdominant males were incubated in seminal fluid from dominant males (*Figure 5*). Contrary to *Cornwallis and O'Connor, 2009*, for which seminal fluid from higher quality ejaculates decreased the velocity of sperm from lower quality ejaculates in fowl, our findings are consistent with the prediction that seminal fluid from ejaculates with faster swimming sperm will enhance the speed of sperm from ejaculates with slower sperm. The disparity between our findings and those in fowl (*Cornwallis and O'Connor, 2009*) possibly reflect differences in the reproductive biology of these species; including internal and external modes of fertilisation and differences in the structure and formation of social hierarchies and associated sperm competition risk.

Ejaculate allocation in fowl is also influenced by factors other than sperm competition risk, including female quality and the probability of future mating opportunities (*Pizzari et al., 2003*; *Cornwallis and Birkhead, 2006*; *Cornwallis and Birkhead, 2007a*; *Cornwallis and Birkhead, 2007b*); whether such factors influence ejaculate allocation strategies in salmonids is unknown. It is also possible that seminal fluid in fowl has evolved to interact with sperm from rivals, as observed in some insect species (*den Boer et al., 2010*) and reported for the grass goby (*Zosterisessor ophiocephalus*) (*Locatello et al., 2013*). Fertilisation occurs rapidly in salmonids, with the majority of eggs fertilised within 10 s post ejaculation (*Hoysak and Liley, 2001*; *Liley et al., 2002*; *Yeates et al., 2007*). Such rapid time frames may allow for little interaction between seminal fluid and sperm from different males during spawning. This is supported by research using Arctic charr that found the activation of sperm with a solution containing seminal fluid from another male had no effect on sperm velocity (*Rudolfsen et al., 2015*). However, a recent experiment that separated and recombined ejaculates from precocious chinook salmon males (obligate sneakers) and adult hooknose males report similar results to those found in the grass goby, with seminal fluid from precocious males significantly decreasing the velocity of hooknose male sperm (*Lewis and Pitcher, 2017*). Our results

**Table 4.** Generalised linear mixed effects models (GLMM) predicting the fertilisation success of male chinook salmon in sperm competition trials using two males and one female.

Trials were conducted using two seminal fluid (SF) treatments, either unmanipulated milt, or recombined ejaculates for which the sperm for both competitors were recombined with the seminal fluid of their rival. Sperm concentration was controlled so that the same number of sperm were used for each male. The first models used the social status of each male to predict the proportion of off-spring sired (n = 20). The second models used the relative sperm velocity (VAP, $\mu$m s$^{-1}$) between competitors to predict the difference in offspring sired (n = 20). The final model shows that the change in relative sperm velocity between males within the same male-male-female combinations across SF treatments was a significant predictor of the change in the proportion of eggs sired by that male's sperm across SF treatments (n = 20). p-Values are calculated using Satterthwaite approximations to degrees of freedom and 95% confidence intervals were calculated using the Wald method. Significant values are highlighted in bold.

| Response variable | SF treatment | Parameters (fixed effects) | Estimate | 95% | P value |
|---|---|---|---|---|---|
| Proportion of offspring sired | Unmanipulated | Intercept<br>Social status | −0.38<br>1.11 | 0.63–1.58 | **<0.0001** |
| | Recombined | Intercept<br>SF social status | −3.24<br>6.23 | 4.7–7.7 | **<0.0001** |
| Difference in number of offspring sired between males | Unmanipulated | Intercept<br>Relative sperm velocity | −1.49e$^{03}$<br>1.44e$^{−01}$ | 0.06–0.23 | **0.003** |
| | Recombined | Intercept<br>Relative sperm velocity | 3.72e$^{03}$<br>0.13 | 0.05–0.21 | **0.003** |
| Difference in proportion of eggs sired across SF treatments | NA | Intercept<br>Difference in relative sperm velocity across SF treatments | −56.39<br>0.006 | 3.5e$^{−03}$– 7.6e$^{−03}$ | **0.0001** |

DOI: https://doi.org/10.7554/eLife.28811.023

The following source data available for Table 4:

Source data 1. Source data for GLMM models predicting the fertilisation success of male chinook salmon in sperm competition trials.

This Excel file contains data on the proportion of eggs sired by each male and the social status of those males. The data is presented in two tabs, the first for the unmanipulated milt and the second for the recombined ejaculate seminal fluid treatments.

DOI: https://doi.org/10.7554/eLife.28811.024

Source data 2. Source data for GLMM models predicting the fertilisation success of male chinook salmon in sperm competition trials.

This Excel file contains data on the diffence in the number of eggs sired between males in each sperm competition trial and the relative sperm velocity of those males. The data is presented in two tabs, the first for the unmanipulated milt and the second for the recombined ejaculate seminal fluid treatments.

DOI: https://doi.org/10.7554/eLife.28811.025

suggest chinook salmon seminal fluid has not evolved a targeted effect on sperm from males adopting a different tactic within the same age/size class, as regardless of social status, males that have faster recorded sperm velocities produced seminal fluid that increases the velocity of sperm from other males with slower speeds, and likewise males with slower sperm velocity produced seminal fluid that decreases the velocity of sperm from males with faster speeds (*Figure 6*).

In addition to demonstrating that seminal fluid influences sperm competitiveness, our *in vitro* sperm competition trials show the influence seminal fluid has on sperm velocity translates to having an effect on male fitness. We measured the fertilisation success within the same male x male x female combinations across trials, and compared those males across unmanipulated and recombined ejaculate treatments, finding changes in the relative sperm velocity between competitors were significantly correlated with the change in the proportion of eggs sired by each male (*Figure 7*). That is, the change in sperm velocity due to the seminal fluid in which sperm were incubated had a significant influence on the proportion of eggs sired by those sperm, in some cases completely reversing the 'winner' of sperm competition within the same male-female group. We now need further investigation to determine the component of seminal fluid that is strategically adjusted by males in response to sperm competition risk.

Previous studies have found that natural variation in several seminal fluid metrics was not correlated with sperm velocity in chinook salmon, including pH, osmolality and ion composition (*Rosengrave et al., 2009a*; *Flannery et al., 2013*). It is possible that seminal fluid contains different levels of available nutrients therefore fuelling differential energy production in sperm. In the short

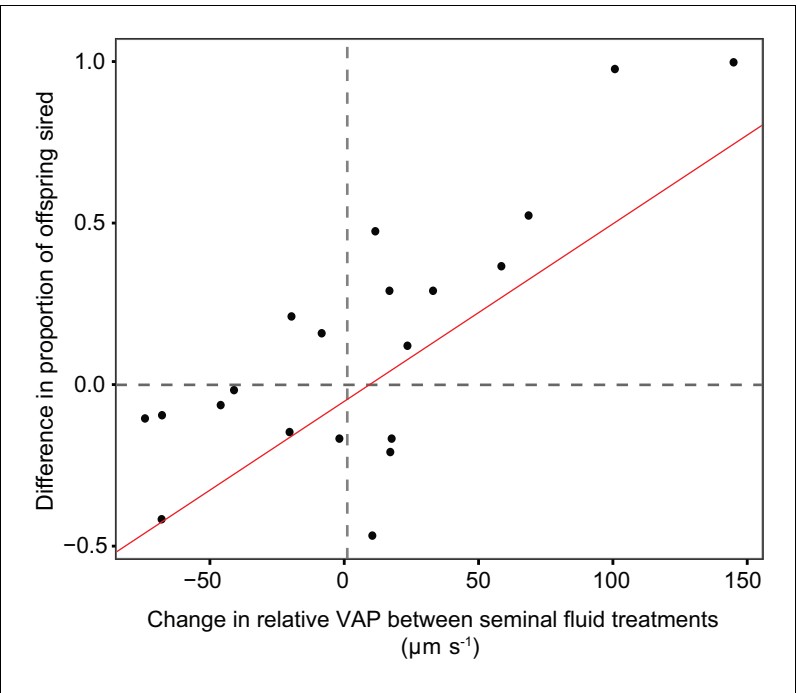

**Figure 7.** Statistically significant relationship between the difference in the proportion of eggs sired by the focal male in each triad from sperm competition trials using chinook salmon (n = 20) when that male's sperm were either incubated in their own or their rival's seminal fluid, and the difference in relative sperm velocity (VAP, μm s$^{-1}$) between males in each pair when sperm were either incubated in their own or their rival's seminal fluid. The relationship shows that change in fertilisation success across seminal fluid treatments is correlated with the change in relative sperm velocity between competing males in each seminal fluid treatment.

DOI: https://doi.org/10.7554/eLife.28811.021

The following source data is available for figure 7:

**Source data 1.** Source data for *Figure 7*.

DOI: https://doi.org/10.7554/eLife.28811.022

term following activation of motility in salmonids, sperm utilise ATP as the energy source for flagellar movement (*Christen et al., 1987*) using both stored ATP reserves and increasing ATP production significantly via aerobic respiration (*Lahnsteiner et al., 1993*, *Lahnsteiner et al., 1999*). Sperm ATP levels have been positively correlated with sperm velocity (*Lahnsteiner et al., 1998*; *Bencic et al., 1999*; *Burness et al., 2004*) and fertilisation success (*Zilli et al., 2004*; *Vladić et al., 2010*) in external fertilisers. Exposure to different levels of exogenous nutrients in seminal fluid while sperm are immotile in the testis may influence energy metabolism, for example altering available energy reserves or stored nutrient reserves, influencing sperm velocity post activation (*Lahnsteiner et al., 1999*). Alternatively, seminal fluid may contain peptide or RNA signalling molecules, that alter sperm behaviour. For example, chemotaxis in several marine invertebrates is controlled by signalling pathways that are initiated by chemoattractant peptides released by ova (*Kaupp et al., 2003*; *Darszon et al., 2008*; *Evans and Sherman, 2013*). Evidence is also accruing that proteins and RNAs in seminal fluid exosomes may play critical roles in regulating sperm development and fertilisation (*Vojtech et al., 2014*; *Jodar et al., 2016*).

Several Seminal Fluid Proteins (SFPs) have been associated with sperm velocity in vertebrate species (*Lahnsteiner et al., 1996*, *1998*; *Poiani, 2006*) and are therefore likely candidates for modifying rapid adjustment of sperm velocity (*Simmons and Fitzpatrick, 2012*). Differences in SFP composition have been documented among males adopting different reproductive tactics in externally fertilising fish (*Scaggiante et al., 1999*; *Gombar et al., 2017*). Additionally, a growing body of empirical work has demonstrated that males can tailor SFP composition in response to sperm competition risk (*Wigby et al., 2009*; *Fedorka et al., 2011*; *Ramm et al., 2015*; *Simmons and Lovegrove, 2017*)

and the mating status of females (*Sirot et al., 2011*). The role of SFPs in sperm competition, with the exception for some insect species (*den Boer et al., 2010*; *Avila et al., 2011*) and specific proteins in mammals (*Ramm et al., 2008*), is generally poorly understood. While the activity of SFPs associated with energy metabolism and respiration have been correlated with sperm velocity in a Cyprinid species (*Lahnsteiner et al., 1996*) and rainbow trout (*O. mykiss*) (*Lahnsteiner et al., 1998*), total protein concentration as well as the activity of lactate dehydrogenase, anti-trypsin and superoxide dismutase enzymes were not correlated with sperm velocity in chinook salmon (*Flannery et al., 2013*). However, these SFPs represents only a small fraction of the enzymatic activity likely to occur in fish seminal fluid (*Gombar et al., 2017*; *Nynca et al., 2014*). The critical next step in determining the molecular mechanism(s) involved will be to link variation in seminal fluid components to sperm velocity, and confirm these results experimentally.

In conclusion, as predicted by sperm competition theory (*Parker, 1990*; *Parker et al., 1997*; *Parker, 1998*; *Wedell et al., 2002*; *Birkhead et al., 2009*; *Parker and Pizzari, 2010*), we find male chinook salmon can make rapid adjustment to sperm velocity in response to social cues that signal changing sperm competition risk and such changes have a significant impact on the outcome of sperm competition and therefore male fitness. We further demonstrate that seminal fluid, even in a species with external fertilisation, plays a key role in mediating the strategic rapid adjustment of sperm velocity and for the first time provide strong evidence the mechanism behind plasticity in sperm velocity lies within the non-sperm component of the ejaculate. Our results support plastic adjustment of ejaculate quality in response to changing sperm competition risk is an effective evolutionary strategy in systems with dynamic social environments and we show seminal fluid mediates such adjustments.

## Materials and methods

### Study species and maintenance

Wild chinook salmon were caught during their annual spawning runs in a trap located on the Kaiapoi River, a tributary of the Waimakariri River system, Canterbury, New Zealand (*Unwin et al., 2000*). We studied a total of 17 sexually mature 3-year-old females and 44 sexually mature 3-year-old 'hooknose' males captured between 27 April and 30 May in 2013, 2014 and 2015. Sample size was informed by related empirical research in this system (*Rosengrave et al., 2008*, *2009a*). Fish were individually tagged and maintained in a natural river-water raceway (12.5–13°C) at a hatchery (Salmon Smolt NZ, Canterbury, New Zealand) using standard husbandry procedures. All animals were collected and maintained according to the standards of the Animal Ethics Committee for the University of Otago, New Zealand.

### Manipulation of male social status

A total of 11 social status manipulation trials were conducted each using four males (n = 44; *Figure 1*). On day 1, two male dyads were formed pairing males of similar size (average fork length = 71.5 cm, 95% CI = 70.2–72.9 cm, n = 44). Each dyad was then placed in a sectioned off part of a river-water raceway (approx. 2.5 m x 2 m x 1 m). Social interactions between the two fish in each dyad were observed for the first day using a series of 10 min under-water video recordings (GoPro Hero 3), one taken each hour over a 5-hr period, with the first recording starting 15 min after introducing fish to the raceway. Male dominance was then determined by calculating a Dominance Index (DI; *Winberg et al., 1991*; *Bailey et al., 2000*; see *Behavioural observations*) using the number of aggressive interactions between males. The male with the higher DI was ranked as dominant (D) and the male with the lower DI as subdominant (S, stage 1 - *Figure 1*). On day 2, male dyads were left undisturbed and male social status within each dyad established on day one typically remained unchanged (*Table 5*). On day 3, male dyads were re-formed placing dominant with dominant and subdominant with subdominant, and a new social hierarchy developed with male social status assigned to each male as described for day one. This forced one fish of each original dyad to change his social status (DS or SD), while the other retained their original status (DD or SS, stage 2 - *Figure 1*). On day 4, the male dyads were left undisturbed, and the experiment was complete on day 5. We determined social status after all the social challenges except in one case where no interaction between males was recorded in the second stage and thus these individuals were excluded from

**Table 5.** The Dominance Index (DI) of the Dominant (D) and Subdominant (S) males in 11 pairings (6 in stage 1 and 5 in stage 2).

In 2013, behavioural observations were conducted twice for each pair, on the day the pair was formed (as in other years) and the next day as a means to assess the stability of social hierarchies. We found that in 10 out of 11 pairs the status of males determined on the first day did not change from on the second day.

| Social status | | D | D | S | S |
|---|---|---|---|---|---|
| Pair | Stage | Day 1 | Day 2 | Day 1 | Day 2 |
| 1 | 1 | 0.844 | 0.739 | 0.155 | 0.26 |
| 2 | 1 | 0.8 | 0.75 | 0.19 | 0.25 |
| 3 | 2 | 0.829 | 0.857 | 0.17 | 0.14 |
| 4 | 2 | 1 | 0.93 | 0 | 0.06 |
| 5 | 1 | 0.98 | 1 | 0.01 | 0 |
| 6 | 1 | 0.96 | 0.89 | 0.03 | 0.1 |
| 7 | 2 | 0.82 | 0.15 | 0.2 | 0.8 |
| 8 | 1 | 0.97 | 1 | 0.03 | 0 |
| 9 | 1 | 1 | 1 | 0 | 0 |
| 10 | 2 | 0.85 | 1 | 0.15 | 0 |
| 11 | 2 | 1 | 1 | 0 | 0 |

DOI: https://doi.org/10.7554/eLife.28811.026

further analyses. A further four males were excluded from analyses due to males escaping from the raceway in the second stage of the experiment, giving a total sample sizes n = 44 in stage 1 and n = 38 in stage 2.

## Behavioural observations

Dominance Index (DI) was calculated using the following equation:

$$DI = Agg^+ / (Agg^+ + Agg^-),$$

where $Agg^+$ represents the total number of aggressive acts performed and $Agg^-$ the total number of aggressive acts received by the individual (*Zilli et al., 2004*; *Bailey et al., 2000*). Aggressive acts were scored using the following criteria:

### Charge
Makes a rapid movement towards the other male.

### Chase
Continual movement towards the other male with that male actively moving away from aggressor. Each lap around the enclosure from the point where the chase was initiated was scored as one chase, such that continual chasing without pause was scored repeatedly.

### Bite
Bites the body of the other male with full gape.

### Nip/Nudge
Bites the tail fin of the other male or nudges the other male with a closed mouth.

## Measurement of ejaculate quality

Ejaculates were obtained from males by gently applying pressure to the abdomen, taking care to avoid contaminating samples, and were held at 4°C for up to 4 hr. We depleted the ejaculate reserves of each male before the experiment so ejaculates collected later were produced during each 48-hr period. We collected ejaculates in a random order on day 3 at the end of stage 1 and

after social status was manipulated on day 5 at the end of stage 2 so samples were collected 48 hr after social status was established in each stage.

Sperm velocity measurements were performed in a random order and blind to the social status of each male. We measured sperm swimming speed twice for each male at 10 s post-activation using a CEROS sperm tracker (v 1.2, Hamilton-Thorne Research, Beverly, MA). Approximately 1 µl of milt was activated with river water or ovarian fluid (diluted to 50% with river water) onto a 20 µl Leja slide (Leja Products B.V., Nieuw-Vennep, The Netherlands) on a temperature-controlled stage cooler (TS-4 Thermal Microscope Stage, Physitemp) set to 12.5°C to match the natural spawning water temperature. We used average path velocity (VAP, µm s$^{-1}$) as our measure of sperm swimming speed which estimates the average velocity of a sperm cell for 0.5 s over a smoothed path (*Rosengrave et al., 2008*, *2009a*, *2016*; *Figure 2—figure supplement 1*). Sperm concentration (sperm/ml) was determined using a Neubauer haemocytometer.

## Manipulation of ejaculates

To determine the relative roles of sperm and seminal fluid on sperm velocity we centrifugally separated and remixed sperm and seminal fluid of each male with those from the other male in each dyad (n = 42 males in 39 dyads). To prepare recombined ejaculates, milt was centrifuged in 1.5 ml tubes at 4°C, 300 g for 10 min to separate sperm cells from seminal fluid. The seminal fluid was then transferred into a new tube after which 500 µl of artificial seminal fluid (80 mM NaCl, 40 mM KCl, 1 mM CaCl$_2$, 20 mM Tris-HCl) was added to the sperm cells and this was centrifuged again at 4°C, 300 g for 10 min to wash any remaining seminal fluid from the sperm cells. The artificial seminal fluid was then discarded and recombined ejaculates were prepared using 10 µl of sperm resuspended in 90 µl of seminal fluid from the same male (control) or seminal fluid from their rival, incubated at 12°C for 20 min.

## *In vitro* fertilisation trials

In 2014 and 2015, at both stages of the social status manipulation trials (*Figure 1*) we conducted a total of 21 replicated *in vitro* fertilisation trials to determine the effects of ejaculate recombination (seminal fluid) on male fertilisation success. This involved 24 individual males and 17 females in which sperm from the dominant and subdominant male in each dyad competed to fertilise a female's eggs. For each trial, we performed two seminal fluid treatments, using either unmanipulated or recombined ejaculates, in addition to non-competitive controls using sperm from each of the males individually. Haphazardly chosen female fish were killed with a stroke to the head, and their egg batch was expelled through an incision in the abdomen, into a clean bowl. Ovarian fluid was collected by carefully pipetting from each egg batch. Sperm density was adjusted prior to each fertilisation trial so that approximately the same number of sperm per male (10$^7$ spermatozoa) were used in each trial.

For each trial, we placed approximately 100 unfertilised ova from the focal female in a dry 2 l plastic beaker, then added ejaculate samples from each male simultaneously by injecting them separately into a steady stream of raceway water (250 ml at 12.5–13 °C). This technique simulated natural spawning conditions by facilitating the rapid mixing of eggs with sperm from both males (*Rosengrave et al., 2016*). We added the ejaculate samples separately into the water to ensure the spermatozoa were activated before the ejaculate samples came into contact, minimising any effects of each male's seminal fluid on the other male's sperm function. The eggs were allowed to sit for 5 min undisturbed until water hardened and were then gently transferred to heath rack trays (12.5–13°C). We randomly sampled 24 alevins from each replicate fertilisation trial (40 days post fertilisation), placing them in 99% ethanol for DNA extraction and microsatellite genotyping to assess paternity.

## DNA extraction, microsatellite amplification and genotyping protocols

To assess paternity share for the males in each sperm competition trial, DNA was extracted from a fin clip for both adult males, the female and 24 offspring from each trial using Chelex100 resin (*Walsh et al., 1991*). Three microsatellite loci (Ots 100, Ots 101, Oki 3a; *Table 6*) were then amplified in a multiplex PCR and used to determine paternity by manually matching alleles between offspring, mother and either potential sire. A fourth locus (Ots 104; *Table 6*) was amplified separately

**Table 6.** Microsatellite primers used to determine paternity.
Primers Ots 100, Ots 101 and Oki 3a were amplified in a multiplex reaction, Ots 104 was amplified singly using a touchdown protocol. Letter at 5' end indicates fluorescent label: p=Pet (red), F = Fam (blue), N = Ned (yellow), V = Vic (green).

| Primer | | Primer sequence 5'−3' | Master mix | PCR | Source |
|---|---|---|---|---|---|
| Ots 100 | F | P-tga-aca-tga-gct-gtg-tga-g | Multiplex | Multiplex | *Nelson and Beacham (1999)* |
| | R | P-acg-gac-gtg-cca-gtg-ag | | | |
| Ots 101 | F | F-acg-tct-gac-ttc-aat-tgg-t | Multiplex | Multiplex | *Small et al. (1998)* |
| | R | F-tat-taa-tcc-tcc-aac-cca-g | | | |
| Oki 3a | F | N-tgt-gct-ata-ggc-tga-atg-tgc | Multiplex | Multiplex | Unpublished, See, *Kinnison et al. (2002)* |
| | R | N-aac-aca-ggc-atc-ccc-act-aa | | | |
| Ots 104 | F | V-gca-ctg-tat-cca-cca-tga | Single | Touchdown | *Nelson and Beacham (1999)* |
| | R | V-gta-gga-gtt-tca-ttt-gaa-tc | | | |

DOI: https://doi.org/10.7554/eLife.28811.027

using a touchdown PCR protocol and employed when three loci were insufficient to determine paternity without certainty. The genotype of each offspring was always consistent with the expected genotype based on the alleles for the potential sires, i.e. in no offspring did we record unique alleles present for both potential sires.

Multiplex PCRs were run in 10 µL volume reactions and included the following reagents: 1x PCR buffer (Bioline), 2 mM $MgCl_2$, 0.3 mM dNTPs, 0.4 µM forward and reverse Ots 101 primers, 0.2 µM forward and reverse Ots 100 and Oki 3a primers, 0.5 U of Bioline Taq DNA polymerase, and 0.5 µL of DNA. The thermal cycling conditions for the multiplex protocol were: 12 min at 95℃ followed by 10 cycles of 15 s at 94℃, 30 s at 60℃, and 30 s at 72℃, followed by 30 cycles of 15 s at 89℃, 30 s at 60℃, 30 s at 72℃, and a final extension period of 10 min at 72℃.

PCRs for amplification of Ots 104 were run in 10 µL volume reactions and included the following reagents: 1x PCR buffer (Bioline), 2 mM $MgCl_2$, 0.3 mM dNTPs, 0.5 µM forward and reverse Ots 104 primers, 0.5 U of Bioline Taq DNA polymerase, and 0.5 µL of DNA. The thermal cycling conditions for the touchdown protocol were: 2 min at 95℃ followed by 10 cycles of 30 s at 95℃, 45 s at Ta℃, and 30 s at 72℃, where Ta starts at 55℃ and drops by 0.5℃ each cycle (last cycle should be 50.5℃), followed by 20 cycles of 30 s at 95℃, 45 s at 50℃, 30 s at 72℃, and a final extension period of 10 min at 72℃.

PCR samples were genotyped by adding 0.5 µL PCR product to 12 µL HiDi formamide and 0.3 µL Genescan LIZ500 size standard (Applied Biosystems) then run on an ABI3130 × 1 Genetic Analyser (Applied Biosystems). Results were visualised using GeneMarker v 2.2 (SoftGenetics, RRID:SCR_015661) and alleles were scored manually.

## Statistical analyses

All statistical analyses were performed using R v 3.1.3 (*R Core Team, 2016*; RRID:SCR_001905). To compare changes in ejaculate quality (sperm velocity (VAP) or sperm concentration) between D and S males, generalised linear mixed effects models (GLMM) were fitted using the package 'lme4' (*Bates et al., 2015*; RRID:SCR_015654). GLMMs using a Gaussian error distribution were fitted using VAP as the response variable, while GLMMs with a Poisson error distribution were fitted using sperm concentration as the response variable. Each GLMM used male social status as a fixed predictor, for stage 1 two levels comparing D and S; and for stage 2, separate models were run with either two levels comparing D and S males with data pooled together (D = D**D** + S**D** and S = S**S** + D**S**), or four levels (males that retained the same status DD and SS, and males that changed status SD and DS). Models with VAP as the response variable used both replicate measurements for each male and included male identity as a random predictor to account for repeated measures.

To test whether males that change social status adjust ejaculate quality, we compared both VAP (GLMMs using a Gaussian error distribution) and sperm concentration (GLMMs with a Poisson error distribution) in the same males across the two stages of the experiment. Four separate models were run for each of the response variables, separately comparing males in each of the four social

phenotypes (DD, DS, SD, SS) and each model used experimental stage (factor with two levels) as a fixed predictor. Additionally, we used an alternative analysis for each of the response variables to test for an interaction effect between social status and experimental stage, both models used social status (factor with four levels; DD, DS, SD and SS), experimental stage (factor with two levels) and the interaction between social status and experimental stage as fixed predictors. Male identity was included as a random predictor to account for repeated measures from the same male.

A linear mixed effects model (GLMM) was fit using the difference in VAP between focal male's sperm recombined with his own seminal fluid and focal male's sperm recombined with his rival male's seminal fluid as the response variable, with difference in VAP between focal male's sperm recombined with his own seminal fluid and rival male's sperm recombined with his own seminal fluid, and social status of rival's seminal fluid as fixed predictors. To fulfil the model's assumption of normality a cube-root transformation was performed on the response variable. We used the random predictors focal male identity, rival male identity and each pairing to account for repeated measures. All VAP measures used were those activated in river water, not ovarian fluid, to avoid female effects on sperm velocity (*Rosengrave et al., 2009b*, *Rosengrave et al., 2016*) that could mask the influence of seminal fluid.

To assess the importance of sperm velocity as a predictor of fertilisation success, we used a GLMM that was fit using the difference in the number of offspring sired between the focal and rival male in each trial as the response variable, with the relative sperm velocity between males as a fixed predictor. To assess social status as a predictor of fertilisation success we used a binomial GLMM that was fit using the proportion of offspring sired by each male as the response variable, with male social status as a fixed predictor in unmanipulated milt trials and the social status of seminal fluid donor as a fixed predictor in swapped seminal fluid trials. In order to assess the influence of seminal fluid on male fertilisation success, we used a GLMM that was fit using the change in the proportion of eggs sired by each focal male across seminal fluid treatments (within the same triad, i.e. within the same male-male-female combination) as the response variable with the change in relative sperm velocity across treatments used as a fixed predictor. For all above models, we used the random predictors focal male identity, rival male identity, female identity and each unique triad to control for repeated measures. We tested for repeatability of replicate trials conducted for each triad (supplementary material: *Statistical analysis and R code*), removing one triad for which the proportion of eggs sired differed significantly between replicates (n = 20). So that sperm velocity in our model reflected conditions during the fertilisation trials, all VAP measures used were those activated in ovarian fluid, as female effects on sperm velocity can influence the outcome of sperm competition in chinook salmon (*Rosengrave et al., 2009b*, *Rosengrave et al., 2016*).

All mentioned models used the week during the spawning season when milt samples were collected as a random predictor to control for potential seasonal effects on milt quality (*Butts et al., 2010*; *Hajirezaee et al., 2010*), and the year fish were collected as a covariate (*Bolker, 2015*). To determine the significance of fixed effects, we present both 95% confidence intervals (CI) calculated using the Wald method, and p values calculated for linear mixed effects models with the package 'lmerTest' (*Kuznetsova et al., 2016*; RRID:SCR_015656) using Satterthwaite approximations to calculate degrees of freedom. Assumptions underlying parametric models were verified using residual plots and Shapiro tests. An alpha value of 0.05 was used to evaluate the significance of P-values and adjusted for multiple tests using the Bonferroni method. Refer to supplementary file: *Statistical analysis and R code*, for all R code used and output from analyses.

## Statistical analysis and R code

Contains all R (*R Core Team, 2016*; RRID:SCR_001905) code, including output and model diagnostics. The following packages were used: lme4 (*Bates et al., 2015*; RRID:SCR_015654), lmerTest (*Kuznetsova et al., 2016*; RRID:SCR_015656), nlme (*Pinheiro et al., 2015*; RRID:SCR_015655), ggplot2 (*Wickham, 2009*; RRID:SCR_014601), lattice (*Sarkar, 2008*; RRID:SCR_015662), RVAideMemoire (*Hervé, 2016*; RRID:SCR_015657), LMERConvenienceFunctions (*Tremblay and Ransijn, 2015*; RRID:SCR_015658), and Deducer (*Fellows, 2012*; RRID:SCR_015659).

## Acknowledgements

We are grateful to the hatchery staff at Salmon Smolt, New Zealand, in particular Karl French, Errol White and Luke Price for providing facilities, fish husbandry, gamete handling assistance and advice. We are also grateful to Fish and Game, New Zealand, in particular Dirk Barr, for invaluable logistical support. We thank Bob Montgomerie and Boris Baer for their invaluable support and discussions regarding experimental design. We also thank Clelia Gasparini for preliminary field assistance and computer-assisted sperm analyses, and Ilina Cubrinovska for field assistance.

## Additional information

### Funding

| Funder | Grant reference number | Author |
| --- | --- | --- |
| Royal Society of New Zealand | Marsden Grant no. UOO1209 | Patrice C Rosengrave |

The funders had no role in study design, data collection and interpretation, or the decision to submit the work for publication.

### Author contributions

Michael J Bartlett, Conceptualization, Data curation, Formal analysis, Validation, Investigation, Visualization, Writing—original draft, Writing—review and editing; Tammy E Steeves, Supervision, Project administration, Writing—review and editing; Neil J Gemmell, Conceptualization, Writing—review and editing; Patrice C Rosengrave, Conceptualization, Supervision, Funding acquisition, Investigation, Methodology, Project administration, Writing—review and editing

### Author ORCIDs

Michael J Bartlett (iD) http://orcid.org/0000-0002-4253-6589
Neil J Gemmell (iD) http://orcid.org/0000-0003-0671-3637
Patrice C Rosengrave (iD) http://orcid.org/0000-0001-8421-3094

### Ethics

Animal experimentation: All animals were collected and maintained according to the approved standards of the Animal Ethics Committee for the University of Otago, New Zealand.

### Decision letter and Author response

Decision letter https://doi.org/10.7554/eLife.28811.033
Author response https://doi.org/10.7554/eLife.28811.034

## Additional files

### Supplementary files

• Supplementary file 1.
DOI: https://doi.org/10.7554/eLife.28811.028

• Transparent reporting form
DOI: https://doi.org/10.7554/eLife.28811.029

### Major datasets

The following dataset was generated:

| Author(s) | Year | Dataset title | Dataset URL | Database, license, and accessibility information |
| --- | --- | --- | --- | --- |
| Bartlett MJ, Steeves TE, Gemmell NJ, Rosengrave PC | 2017 | Data from: Sperm competition risk drives rapid ejaculate adjustments mediated by seminal fluid | https://doi.org/10.5061/dryad.kr137 | Available at Dryad Digital Repository under a CC0 Public |

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
