## [Decision Letter]

Thank you for submitting your article "Sperm competition risk drives rapid ejaculate adjustments mediated by seminal fluid" for consideration by *eLife*. Your article has been reviewed by three peer reviewers, and the evaluation has been overseen by a Reviewing Editor and Ian Baldwin as the Senior Editor. The following individuals involved in review of your submission have agreed to reveal their identity: Matthew Gage (reviewer #1); Tom Pizzari (reviewer #2).

The reviewers have discussed the reviews with one another and the Reviewing Editor has drafted this decision to help you prepare a revised submission.

Summary:

Two independent reviews have now been submitted for this manuscript; one is extremely positive, and the other contains more caveats. The less positive reviewer sees value in the work, but concludes that general presentation of the manuscript and referencing to previous work needs further work. Both reviewers suggest a number of specific routes to improvement, which we see as being easily surmountable.

This is a strong and valuable piece of work that could easily be improved to a very high standard by addressing both reviewers' comments. After this, the work would constitute a major contribution to our understanding in reproductive biology and evolution.

The work combines a series of complementary findings within one study that sum to an important advance. There are three well-executed experimental manipulations which allow clear findings that: a) male dominance status influences sperm behaviour and this can change over short time periods; b) seminal fluid biochemistry is a primary driver behind this change in sperm behaviour; and c) the change in sperm behaviour as a consequence of seminal fluid biochemistry translates into male reproductive success. From this work, which uses an ideal model system, we can conclude that males are sensitive to proximate risks of competition for fertilisations, and they modulate sperm function over short time periods to match the risk of competition using seminal fluid. Altogether, these are important findings that allow novel insights into the surprising extents to which males and their sperm are shaped by the struggle to reproduce.

Essential revisions:

The authors should refer to the two separate reviews in order to satisfy their concerns about the current state of the manuscript.

The more 'problematic' issues raised by reviewer 3 are also readily addressed as they concern originality, presentation style, and statistics.

In reverse order: reviewer 3 asks for clarity concerning statistical analysis of the switch in social status and effects on VAP, clarity which would be helpful for readers.

In terms of presentation, it would be helpful to use original references that reviewer 3 is concerned about, and to flesh out some of the theory on sperm competition.

Considering originality, which links to the presentation and overstatement concerns of reviewer 3: I agree that Locatello et al., 2013 work with goby males also used the clever transfer of sperm and seminal fluid between morphs to show that fluid can indeed influence sperm function, however the current study did refer to this work in the introduction. Despite this study, conclusive empirical findings of the consequence of seminal fluid changes, and that these can happen in the short term in relation to dominance status have been so far non-existent. I therefore see this as the strength of the current work, and it would strengthen the manuscript to ensure both Locatello et al., and Scaggiante et al., studies are mentioned clearly in the Introduction.

Finally, reviewer 3 is concerned that the seminal fluid compositional changes were not measured in tandem with sperm behaviour changes and the consequences for fertilisations success. I believe, with reviewer 2, that this is beyond the scope of the current work, and may be a huge task given the diversity of compounds within male seminal fluids. However, I believe it would strengthen the manuscript to include wider discussion of how seminal fluid might influence sperm behaviour, as suggested by reviewer 3. An additional thought to consider is whether seminal fluid acts as a signal for sperm behavioural switching, or whether it contains exogenous nutrients that 'fed' and then changed sperm motility? This can be considered in the light of what is known about exogenous and endogenous metabolism in the sperm of external fertilisers. I believe the consensus is that there is rather little evidence for metabolism of exogenous nutrients in the short term following spermatozoal activation – but it may be different following incubation. It would be worth considering a brief discussion on this point, including reference to reviewer 3's question about possible RNA signalling, and the finding of glycoprotein variation in goby seminal fluid composition (Scaggiante et al., 1999). It also relates to the strong work on fowl where rival male ejaculates have had different and interesting impacts on sperm behaviour under male dominance variation. If seminal fluid contains signals, for example, which sperm react to, then the shift in velocity may well trade against other activity components such as longevity, with changes to behaviour possibly influenced by both signaller and receiver. If, on the other hand, seminal fluid contains nutrients for exogenous metabolism by sperm, then some discussion about sugars/glycoproteins as well as SFPs would be helpful.

Reviewer #2:

This study investigates experimentally: (a) the influence of seminal fluid in patterns of variation in sperm swimming velocity, a key driver of fertilising efficiency, and (b) the extent to which such seminal fluid-driven changes in sperm swimming velocity impact the outcome of sperm competition. The possibility that seminal fluid compounds, primarily accessory gland products, influence the competitive efficiency of an ejaculate, and have largely evolved for their role in sperm competition, has been around for decades but direct evidence in support of this has been lacking. This is not surprising given the technical difficulties involved in testing this idea under controlled experimental conditions (as way of reference, my own research group has been working on these very questions and using similar approaches for years now, and we haven't yet submitted for publication because our data are not yet as clear and as complete as these presented here). The present study of a chinook salmon population presents so far the most convincing piece of evidence that I have seen in support of the notion that seminal fluid drives changes in sperm fertilising efficiency with tangible impact on sperm competition. Importantly, these results also suggests that males are able to strategically change the composition of their seminal fluid in order to respond to changes in the risk of sperm competition associated with rapid changes in social status. Specifically, by presenting males with a double social challenge the study shows that males that lost the first contest (SS + SD) produce faster swimming sperm than males that won the first contest (DD + DS), and that -after controlling for multiple comparisons- the sperm of males that decreased in status over the two challenges (DS) increased in swimming velocity. Given that these changes occurred rapidly (~48hrs) and males were experimentally sperm depleted, it is parsimonious to assume that they are due to seminal fluid changes rather than sperm phenotype. This is then confirmed by a subsequent experiment in which the sperm of a focal male is incubated in the seminal fluid of a competitor male, and the change in the focal male's sperm velocity is then shown to be predicted by the social status of the competitor male. Critically, when the relative sperm swimming velocity of the competitor male is entered in the model, this becomes the significant predictor while status becomes non-significant, suggesting that male status explains changes in the sperm velocity only to the extent to which it predicts the relative swimming velocity of the competitor male's sperm. This demonstrates that the seminal fluid of the competitor male has a causal effect in the swimming velocity of the focal male. Finally, the study performs a series of in vitro competitive fertilisation trials to show that when the sperm of a focal male are exposed to the seminal fluid of his respective rival, changes in fertilisation success are best predicted by the changes in sperm swimming velocity due to exposure to the rival's seminal fluid. Together, these results present a robust case for socially subordinate males compensating through preferential investment in seminal fluid components which increase the fertilising efficiency of their ejaculates, confirming previous suggestions of alternative mating tactics in this and other fish species with similar mating systems. Overall, I think that this is an excellent study. The experimental work is conducted meticulously and impressive in scale. The manuscript is written clearly and with appropriate consideration of previous work. The conclusions drawn are justified and discussed in adequate depth.

Reviewer #3:

This study investigates how male social status (i.e. dominant versus subordinate) affects sperm performance in male chinook salmon. They paired males up and let them establish a hierarchy and collected sperm after 48 hours, male were then re-paired by social status (dominant with dominant and subordinate with subordinate) forcing half of the ales to switch status. The authors tested how seminal fluid affects sperm performance by separating sperm from seminal fluid and re-suspending sperm in the seminal fluid of a different male. They found subordinate males to produce generally faster swimming sperm than dominant males. Furthermore, they show that sperm from dominant males in subordinate seminal fluid became faster and vice versa and that the resulting sperm velocity was a strong predictor of fertilisation success.

While the study is solid and results seem convincing (although more statistics are needed), the manuscript could be improved by tightening the writing style and making the arguments and more concrete. The writing style is often loose and ambivalent, which this is also true for the methods section. In addition, the use of references is not always satisfying as it would be great to see references to original papers rather than reviews for example when introducing the theoretical predictions about sperm competition. In general, the manuscript would benefit from careful revision to make it tighter and more accurate.

For one, the authors claim that they have been using a novel experimental approach to disentangle the effect of sperm versus seminal fluid on sperm velocity. However, the same approach has been used in a study by Locatello et al., (2013) published a few years ago showing a similar effect of seminal fluid on sperm performance between males adopting alternative mating tactics in a goby. Furthermore, previous studies showed that sneaker and territorial male ejaculates in a Mediterranean goby differ in the composition and amount of polysaccharides present in the seminal fluid (e.g. Scaggiante et al., 1999). The authors should mention these studies in the Introduction (and not only in the Discussion) and refer to these previous results when introducing their current study. In other words, a more accurate coverage of the literature in the relevant places would help to put this present study into context.

Furthermore, the authors argue that sperm velocity is affected by a change in composition in the seminal fluid based on the fact that males changing from dominant to subordinate did not produce significantly more sperm over the span of 48 hours. Unfortunately, they performed no test – not even a simple test for polysaccharides or general protein content – to actually verify a change in composition of the seminal fluid. There are several alternative explanations for a change in sperm swimming velocity, such as RNAs injected into sperm through vesicles immediately prior to ejaculate release that alter sperm metabolic pathways to name but one, which should also be discussed.

Finally, the fact that males experience a rapid turnover in treatments where males are kept in each social status for three days in total needs a bit more attention. The sequence of the social status clearly has an effect on sperm traits as the main effect on VAP was found in males switching from dominant to subordinate. I am therefore wondering how these issues were taken into account when running the statistical analyses. The authors write that they used either two levels (D and S) when analysing the stages separately or four when analysing stage 2. This is an ambivalent description and should be clarified; for example in Table 3, it is not clear how the statistical model looked like and no interaction terms are presented whereas Figure 4 suggests a clear interaction between treatments (DD and the other three). I am generally wondering about the role of interaction terms between the social status during stage 1 and stage 2 as none are presented in the current version of the manuscript.

In summary, while I can see the value of this study, the general presentation should be improved and the novelty of the findings is overstated.

---

## [Author Response]

Essential revisions:The authors should refer to the two separate reviews in order to satisfy their concerns about the current state of the manuscript.The more 'problematic' issues raised by reviewer 3 are also readily addressed as they concern originality, presentation style, and statistics.In reverse order: reviewer 3 asks for clarity concerning statistical analysis of the switch in social status and effects on VAP, clarity which would be helpful for readers.In terms of presentation, it would be helpful to use original references that reviewer 3 is concerned about, and to flesh out some of the theory on sperm competition.

We have expanded on our explanation of sperm competition theory and include additional empirical references, including highly relevant papers by Parker and colleagues that originally explored sperm competition modelling under different scenarios (Introduction).

Considering originality, which links to the presentation and overstatement concerns of reviewer 3: I agree that Locatello et al., 2013 work with goby males also used the clever transfer of sperm and seminal fluid between morphs to show that fluid can indeed influence sperm function, however the current study did refer to this work in the introduction. Despite this study, conclusive empirical findings of the consequence of seminal fluid changes, and that these can happen in the short term in relation to dominance status have been so far non-existent. I therefore see this as the strength of the current work, and it would strengthen the manuscript to ensure both Locatello et al., and Scaggiante et al., studies are mentioned clearly in the Introduction.Finally, reviewer 3 is concerned that the seminal fluid compositional changes were not measured in tandem with sperm behaviour changes and the consequences for fertilisations success. I believe, with reviewer 2, that this is beyond the scope of the current work, and may be a huge task given the diversity of compounds within male seminal fluids. However, I believe it would strengthen the manuscript to include wider discussion of how seminal fluid might influence sperm behaviour, as suggested by reviewer 3. An additional thought to consider is whether seminal fluid acts as a signal for sperm behavioural switching, or whether it contains exogenous nutrients that 'fed' and then changed sperm motility? This can be considered in the light of what is known about exogenous and endogenous metabolism in the sperm of external fertilisers. I believe the consensus is that there is rather little evidence for metabolism of exogenous nutrients in the short term following spermatozoal activation – but it may be different following incubation. It would be worth considering a brief discussion on this point, including reference to reviewer 3's question about possible RNA signalling, and the finding of glycoprotein variation in goby seminal fluid composition (Scaggiante et al., 1999). It also relates to the strong work on fowl where rival male ejaculates have had different and interesting impacts on sperm behaviour under male dominance variation. If seminal fluid contains signals, for example, which sperm react to, then the shift in velocity may well trade against other activity components such as longevity, with changes to behaviour possibly influenced by both signaller and receiver. If, on the other hand, seminal fluid contains nutrients for exogenous metabolism by sperm, then some discussion about sugars/glycoproteins as well as SFPs would be helpful.Reviewer #2:This study investigates experimentally: (a) the influence of seminal fluid in patterns of variation in sperm swimming velocity, a key driver of fertilising efficiency, and (b) the extent to which such seminal fluid-driven changes in sperm swimming velocity impact the outcome of sperm competition. The possibility that seminal fluid compounds, primarily accessory gland products, influence the competitive efficiency of an ejaculate, and have largely evolved for their role in sperm competition, has been around for decades but direct evidence in support of this has been lacking. This is not surprising given the technical difficulties involved in testing this idea under controlled experimental conditions (as way of reference, my own research group has been working on these very questions and using similar approaches for years now, and we haven't yet submitted for publication because our data are not yet as clear and as complete as these presented here). The present study of a chinook salmon population presents so far the most convincing piece of evidence that I have seen in support of the notion that seminal fluid drives changes in sperm fertilising efficiency with tangible impact on sperm competition. Importantly, these results also suggests that males are able to strategically change the composition of their seminal fluid in order to respond to changes in the risk of sperm competition associated with rapid changes in social status. Specifically, by presenting males with a double social challenge the study shows that males that lost the first contest (SS + SD) produce faster swimming sperm than males that won the first contest (DD + DS), and that -after controlling for multiple comparisons- the sperm of males that decreased in status over the two challenges (DS) increased in swimming velocity. Given that these changes occurred rapidly (~48hrs) and males were experimentally sperm depleted, it is parsimonious to assume that they are due to seminal fluid changes rather than sperm phenotype. This is then confirmed by a subsequent experiment in which the sperm of a focal male is incubated in the seminal fluid of a competitor male, and the change in the focal male's sperm velocity is then shown to be predicted by the social status of the competitor male. Critically, when the relative sperm swimming velocity of the competitor male is entered in the model, this becomes the significant predictor while status becomes non-significant, suggesting that male status explains changes in the sperm velocity only to the extent to which it predicts the relative swimming velocity of the competitor male's sperm. This demonstrates that the seminal fluid of the competitor male has a causal effect in the swimming velocity of the focal male. Finally, the study performs a series of in vitro competitive fertilisation trials to show that when the sperm of a focal male are exposed to the seminal fluid of his respective rival, changes in fertilisation success are best predicted by the changes in sperm swimming velocity due to exposure to the rival's seminal fluid. Together, these results present a robust case for socially subordinate males compensating through preferential investment in seminal fluid components which increase the fertilising efficiency of their ejaculates, confirming previous suggestions of alternative mating tactics in this and other fish species with similar mating systems. Overall, I think that this is an excellent study. The experimental work is conducted meticulously and impressive in scale. The manuscript is written clearly and with appropriate consideration of previous work. The conclusions drawn are justified and discussed in adequate depth.Reviewer #3:This study investigates how male social status (i.e. dominant versus subordinate) affects sperm performance in male chinook salmon. They paired males up and let them establish a hierarchy and collected sperm after 48 hours, male were then re-paired by social status (dominant with dominant and subordinate with subordinate) forcing half of the ales to switch status. The authors tested how seminal fluid affects sperm performance by separating sperm from seminal fluid and re-suspending sperm in the seminal fluid of a different male. They found subordinate males to produce generally faster swimming sperm than dominant males. Furthermore, they show that sperm from dominant males in subordinate seminal fluid became faster and vice versa and that the resulting sperm velocity was a strong predictor of fertilisation success.While the study is solid and results seem convincing (although more statistics are needed), the manuscript could be improved by tightening the writing style and making the arguments and more concrete. The writing style is often loose and ambivalent, which this is also true for the methods section. In addition, the use of references is not always satisfying as it would be great to see references to original papers rather than reviews for example when introducing the theoretical predictions about sperm competition. In general, the manuscript would benefit from careful revision to make it tighter and more accurate.

We have carefully revised our manuscript in accord with the constructive criticisms received. We are confident that it is now stronger as a consequence of the review process and we thank the Editor and all the reviewers for their insightful feedback.

For one, the authors claim that they have been using a novel experimental approach to disentangle the effect of sperm versus seminal fluid on sperm velocity. However, the same approach has been used in a study by Locatello et al., (2013) published a few years ago showing a similar effect of seminal fluid on sperm performance between males adopting alternative mating tactics in a goby. Furthermore, previous studies showed that sneaker and territorial male ejaculates in a Mediterranean goby differ in the composition and amount of polysaccharides present in the seminal fluid (e.g. Scaggiante et al.,). The authors should mention these studies in the Introduction (and not only in the Discussion) and refer to these previous results when introducing their current study. In other words, a more accurate coverage of the literature in the relevant places would help to put this present study into context.

We have now included a clear explanation of Locatello et al., (2013) and Scaggiante et al., (1999) results (Introduction). We have also changed the description of our experimental approach from “innovative” to “comprehensive” (Introduction) to reflect that while each of the methods used here has been previously implemented singularly or partially in other study systems, we have been able to gain novel insight by using these methods in a comprehensive series of experiments in an ideal study system. We have also expanded our discussion of targeted seminal fluid effects on rival male sperm to refer to some recent relevant research in salmonids on this topic (Discussion).

Furthermore, the authors argue that sperm velocity is affected by a change in composition in the seminal fluid based on the fact that males changing from dominant to subordinate did not produce significantly more sperm over the span of 48 hours. Unfortunately, they performed no test – not even a simple test for polysaccharides or general protein content – to actually verify a change in composition of the seminal fluid. There are several alternative explanations for a change in sperm swimming velocity, such as RNAs injected into sperm through vesicles immediately prior to ejaculate release that alter sperm metabolic pathways to name but one, which should also be discussed.

We agree that a wider discussion of how seminal fluid might influence sperm behaviour strengthens our manuscript. We have expanded our discussion on this topic, addressing the potential for seminal fluid to fuel differential energy metabolism in sperm, and the potential for seminal fluid to contain signalling molecules (either RNAs or peptides) that influence sperm behaviour (Discussion). We have also referred to Scaggiante et al., (1999) and a recent study by Gombar et al., (2017) with respect to variation in SFP composition in males with different fixed reproductive tactics (Discussion). Finally, we adjusted our focus on SFPs at the end of this discussion to a broader statement (Discussion).

Finally, the fact that males experience a rapid turnover in treatments where males are kept in each social status for three days in total needs a bit more attention. The sequence of the social status clearly has an effect on sperm traits as the main effect on VAP was found in males switching from dominant to subordinate. I am therefore wondering how these issues were taken into account when running the statistical analyses. The authors write that they used either two levels (D and S) when analyzing the stages separately or four when analyzing stage 2. This is an ambivalent description and should be clarified; for example, in Table 3 it is not clear how the statistical model looked like and no interaction terms are presented whereas Figure 4 suggests a clear interaction between treatments (DD and the other three). I am generally wondering about the role of interaction terms between the social status during stage 1 and stage 2 as none are presented in the current version of the manuscript.

As stated above for reviewer 2’s first comment, we have clarified the methods used for group comparisons in stage 2 (Materials and methods). We have also clarified the methods used for the statistical analysis of change in sperm velocity and sperm concentration from stage 1 to 2 (Materials and methods). Our original analyses used separate models for each of the four social status groups (DD,SD,DS,SS) and simply compared sperm velocity or concentration between stages, with no ability to test for an interaction effect. We have now included an additional analyses for both sperm velocity and concentration in which a single model is run for each response variable with data for all four social status groups to test for an interaction effect between social group and stage. We found that the results for this additional analysis were equivalent to those from our previous analyses and have included them in the manuscript (Results; Materials and methods). We have also updated the supplementary file to include the R code and output for these new analyses.

In summary, while I can see the value of this study, the general presentation should be improved and the novelty of the findings is overstated.